# Specific proteolysis mediated by a p97-directed proteolysis-targeting chimera (p97-PROTAC)

Constanza Salinas-Rebolledo[1], Javier Blesa[2], Guillermo Valenzuela-Nieto[3], David Schwefel[4], Natalia López-González del Rey[2], Maxs Méndez-Ruette[5], Janine Burkhalter[1], Elizabeth Carrazana[6], Francisca Díaz-Tejeda[1], Ignacio Arias Catalán[7], Claudio Cappelli Leon[7], Natalia Salvadores[6], Luis Federico Bátiz[8,9], Ronald Jara[1], José A Obeso[2], Pedro Chana-Cuevas[10], Gopal P Sapkota[11], Alejandro Rojas-Fernandez[1,12,13]*

[1]Institute of Medicine, Faculty of Medicine, Universidad Austral de Chile, Valdivia, Chile; [2]HM CINAC (Centro Integral de Neurociencias Abarca Campal), Hospital Universitario HM Puerta del Sur, HM Hospitales, Madrid, Spain; [3]Biotechnological Exploration Laboratory, Health Care Science Faculty, Universidad San Sebastian, Valdivia, Chile; [4]Technische Universität Berlin, Chair of Bioanalytics, Berlin, Germany; [5]Programa de Doctorado en Biomedicina, Facultad de Medicina, Universidad de los Andes, Santiago, Chile; [6]Neurodegenerative Diseases Laboratory, Center for Biomedicine, Universidad Mayor, Temuco, Chile; [7]Immunoepigenetics Laboratory, Institute of Biochemistry and Microbiology, Faculty of Sciences, Universidad Austral de Chile, Valdivia, Chile; [8]Neuroscience Program, Centro de Investigación e Innovación Biomédica (CiiB) & School of Medicine, Facultad de Medicina, Universidad de los Andes, Santiago, Chile; [9]IMPACT, Center of Interventional Medicine for Precision and Advanced Cellular Therapy, Santiago, Chile; [10]CETRAM & Faculty of Medical Science Universidad de Santiago de Chile, Santiago, Chile; [11]Medical Research Council Protein Phosphorylation and Ubiquitylation Unit, School of Life Sciences, University of Dundee, Sir James Black Centre, Dundee, United Kingdom; [12]Berking Biotechnology SpA, Valdivia, Chile; [13]Berking Bioscience GmbH, Hamburg, Germany

*For correspondence: alejandro.rojas@uach.cl

**Abstract** The p97 protein is a member of the AAA+ family of ATPases. This protein is encoded by the *VCP* gene. It is a mechanoenzyme that uses energy from ATP hydrolysis to promote protein unfolding and segregation actively. The unfolded products are subsequently presented to the 26S proteasome for degradation. p97 substrate recognition is mediated by adaptors, which interact with substrates directly or indirectly through ubiquitin modifications, resulting in substrate funnelling into the central pore of the p97 hexamer and unfolding. Here, we engineered synthetic adaptors to target specific substrates to p97, using the extraordinary intracellular binding capabilities of camelid nanobodies fused to the UBX domain of the p97 adaptor protein Fas-associated factor-1 (FAF1). In such a way, we created a p97-directed proteolysis-targeting chimera (PROTAC), representing a novel and unique E3 ubiquitin ligase-independent strategy to promote specific proteolysis. All functional assays were performed in human cell lines to evaluate the system's efficacy and specificity in a physiologically relevant context.

## Editor's evaluation

This study describes a valuable PROTAC approach in which the UBX domain of FAF1 fused to a nanobody (Ubx-Nb) can target protein of interest for degradation. The authors provide convincing evidence, showing that Ubx-Nb with a nanobody recognizing GFP can reduce the cellular levels of several GFP-fusion model substrates including some aggregation-prone proteins relevant to neuro-degenerative diseases. The study will be of broad interest to cell biologists in the targeted protein degradation field.

## Introduction

The ubiquitin-proteasome system (UPS) plays a central role in maintaining proteostasis by regulating protein abundance. This regulation is mediated by specific E3 ubiquitin ligases, which catalyze ubiquitin chain formation on the substrates, inducing their proteasome-mediated degradation (*Tara et al., 1974*; *Ciechanover, 1994*; *McNaught et al., 2001*; *Schmidt et al., 2021*). The UPS as an efficient natural negative-regulatory mechanism has inspired the development of the proteolysis-targeting chimera (PROTAC) technology, involving synthetic heterobifunctional molecules able to recruit a protein of interest (POI) to a ubiquitin E3 ligase to induce its proteasomal degradation (*Bondeson et al., 2015*; *Sakamoto et al., 2001*; *Holland et al., 2012*; *Röth et al., 2020*; *Macartney et al., 2017*; *Clift et al., 2018*; *Békés et al., 2022*; *Tomoshige et al., 2017*; *Tomoshige and Ishikawa, 2021*; *Pankey, 1986*; *Ibrahim et al., 2020*; *Paiva and Crews, 2019*; *Wang et al., 2020*; *Békés et al., 2022*).

The nature of PROTACs varies from small molecules to protein domains and antibody fragments.

Several camelid nanobodies are capable of binding intracellular target proteins selectively, with a high affinity. When nanobodies are overexpressed in mammalian cells, they are also known as intrabodies (*Ibrahim et al., 2020*; *Caussinus et al., 2011*; *Fulcher et al., 2016*). Nanobodies fused to ubiquitin E3 ligase substrate receptors trigger protein degradation of ectopic and endogenous targets. For instance, the AdPROM system consisting of a fusion of the E3 substrate receptor VHL with nanobodies against a range of POIs leads to the efficient degradation of endogenous POI targets (*Röth et al., 2020*; *Macartney et al., 2017*; *Fulcher et al., 2016*; *Fulcher et al., 2017*; *Simpson et al., 2020*; *Ottis et al., 2017*; *Smith et al., 2019*). Nanobodies have also been engineered and fused directly to the active domains of ubiquitin E3 ligases, such as the antibody RING-mediated destruction system (ARMeD), which uses the RING finger domain of the ubiquitin E3 ligase RNF4 fused to a nanobody. A valuable feature of the ARMeD system is its independence of the endogenous ubiquitin E3 ligases (*Ibrahim et al., 2020*).

In addition to conventional ubiquitin-dependent degradation, the AAA-type ATPase p97 assists the proteasome in the specific selection of substrate degradation in eukaryotic cells (*van den Boom and Meyer, 2018*). The key mechanism of action involves the disassembly of protein complexes through its ATP-dependent *segregase* and *unfoldase* activity (*Noi et al., 2013*; *Olszewski et al., 2019*; *Bodnar and Rapoport, 2017*; *Hu et al., 2020*). p97 uses ATP hydrolysis as a source of energy to 'segregate' ubiquitylated protein complex subunits from their binding partners, or even to the unfolding of protein aggregates, such as tau amyloid fibers and exon 1 of Huntingtin (HTT) (*Wentink and Rosenzweig, 2023*; *Mukkavalli et al., 2021*; *Saha et al., 2023*; *Ghosh et al., 2018*). This action is mediated by two types of adaptors: the UBX-like domain (UBX-L, also known as the 'ubiquitin-binding domain' [UBD]) adaptors such as Ufd1, NLP4, p47, FAF1, SAKS, UBXD7, and UBXD8, and the UBX-only adaptors, such as p37, UBXD1, UBXD2, UBXD3, UBXD4, UBXD5, UBXD6, VCIP135, and YOD1 (*Ye et al., 2017*; *Yeung et al., 2008*; *Kloppsteck et al., 2012*; *Meyer, 2012*; *Stach and Freemont, 2017*).

p97 also regulates ER-mitochondrial association by disassembling Mfn2 complexes upon PINK/Parkin phosphoubiquitination and the mitochondrial extraction of MARCH5. This process disrupts mitochondria-ER tethering during mitophagy, facilitating mitochondrial degradation (*McLelland et al., 2018*; *Koyano et al., 2019*). Additionally, p97 co-localizes with protein aggregates involved in several neurodegenerative diseases, suggesting that its *segregase* and *unfoldase* activities could be related to the proteolytic control of proteins of therapeutic interest (*Galindo-Moreno et al., 2017*; *Hirabayashi et al., 2001*; *Mizuno et al., 2003*; *Yang and Hu, 2016*; *Kobayashi et al., 2007*; *Alieva et al., 2020*). For instance, p97 is recruited to poly-Q aggregates in vitro and to inclusion-positive neurons in Huntington's disease patients (*Mukkavalli et al., 2021*; *Hirabayashi et al., 2001*). Abnormal protein

aggregation is observed in several pathologies such as inclusion body myopathy associated with Paget's disease of bone and frontotemporal dementia (IBMPFD), Charcot-Marie-Tooth disease, amyotrophic lateral sclerosis, and Parkinson's disease (*Alieva et al., 2020*; *Watts et al., 2004*; *Falcão de Campos and de Carvalho, 2019*; *Gonzalez et al., 2014*; *González-Pérez et al., 2012*). Ubiquitin is often found as a resident protein within aggregates related to neurodegenerative diseases, suggesting potential dysfunction of ubiquitin-mediated degradation signaling (*Alnot and Frajman, 1992*; *Huang and Figueiredo-Pereira, 2010*; *Donaldson et al., 2003*; *Fernández-Sáiz and Buchberger, 2010*).

Motivated by these characteristics, we engineered a synthetic p97 adapter by fusing the UBX domain of the FAF1 protein to camelid nanobodies, to assemble a novel p97-based PROTAC (**p97-PROTAC**) technology. This new chimera efficiently targets proteins for segregation and proteasome-mediated degradation in a ubiquitin-independent manner.

## Results
### p97-PROTAC subcellular recruitment and activity in cells

p97 adaptors and cofactors lead to the specific recognition of ubiquitin-modified and unmodified substrates, enabling their simultaneous entry into the central pore of p97 hexamers, followed by unfolding and subsequent proteasomal degradation (*Caffrey et al., 2021*; *Bodnar and Rapoport, 2017*). We hypothesized that p97 adaptors could be engineered to target non-natural substrates of clinical interest for degradation. Thus, we generated a synthetic chimera consisting of the UBX domain of the p97 adapter FAF1 fused via a linker to a GFP-specific camelid nanobody (UBX-Nb(GFP)), which is capable of recognizing both GFP- and yellow fluorescent protein (YFP)-tagged proteins, to yield a novel PROTAC based on p97 activity (*Figure 1A*; amino acid sequence in *Figure 1—figure supplement 1A*).

Using fluorescence microscopy, we then evaluated if the UBX-Nb(GFP) could recognize GFP-fusion proteins with different subnuclear localizations by analyzing its recruitment to a group of differentially located targets: (i) GFP-Coilin (nuclear and Cajal bodies), (ii) GFP-Emerin (a type II integral membrane protein residing principally at the inner nuclear membrane, with occasional reports of ER localization likely due to overexpression or cell-type specific contexts) (*Xie et al., 2024*), and (ii) GFP-ETV1 (nuclear transcription factor) in HeLa cells. Our results demonstrated efficient recruitment of UBX-Nb(GFP) to these diverse nuclear locations (*Figure 1B*). Additionally, we confirmed its colocalization using the Pearson correlation coefficient. High Pearson correlation values in our analysis further support the specific colocalization of UBX-Nb(GFP) with these nuclear proteins. Furthermore, we generated histograms representing the fluorescence intensity profiles of UBX-Nb(GFP) and the target proteins across selected regions of interest, providing a visual confirmation of their spatial overlap (*Figure 1—figure supplement 1B–D*).

To further assess whether the UBX-Nb(GFP) construct exhibits any intrinsic subcellular localization pattern, we co-expressed Myc-UBX-Nb(GFP) together with either free GFP or an empty vector. Under both conditions, UBX-Nb(GFP) displayed a diffuse cytoplasmic and nuclear distribution without enrichment in specific organelles (*Figure 1—figure supplement 1H and I*). These results confirm that the localization of UBX-Nb(GFP) observed in *Figure 1B* is due to its recruitment to GFP-tagged target proteins, and not to an inherent localization bias of the construct.

Next, we studied the effect of Myc-tagged UBX-Nb(GFP) PROTAC expression on the levels of GFP-Coilin, GFP-Emerin, and GFP-ETV1. We observed a significant reduction in the levels of all GFP-tagged target proteins with increasing expression of Myc-UBX-Nb(GFP), indicating that the p97-PROTAC-UBX-Nb(GFP) is sufficient to trigger specific proteolysis of target proteins (*Figure 1C–H*). Consistently, no degradation was observed when the GFP nanobody was replaced by a negative control nanobody (*Figure 1—figure supplement 1E–G*). Therefore, these findings suggested that p97-PROTAC could be an alternative targeted protein degradation approach to the conventional ubiquitin ligase-based PROTACs and might be useful for the degradation of large protein complexes, integral membrane protein at the inner nuclear membrane, and toxic aggregates due to its *unfoldase* and *segregase* activities.

### Specific degradation of proteins within liquid-liquid phase separation structures

We sought to test the efficacy of the p97-PROTAC system for the degradation of proteins whose expression was driven by endogenous promoters. To further investigate its ability to induce degradation of

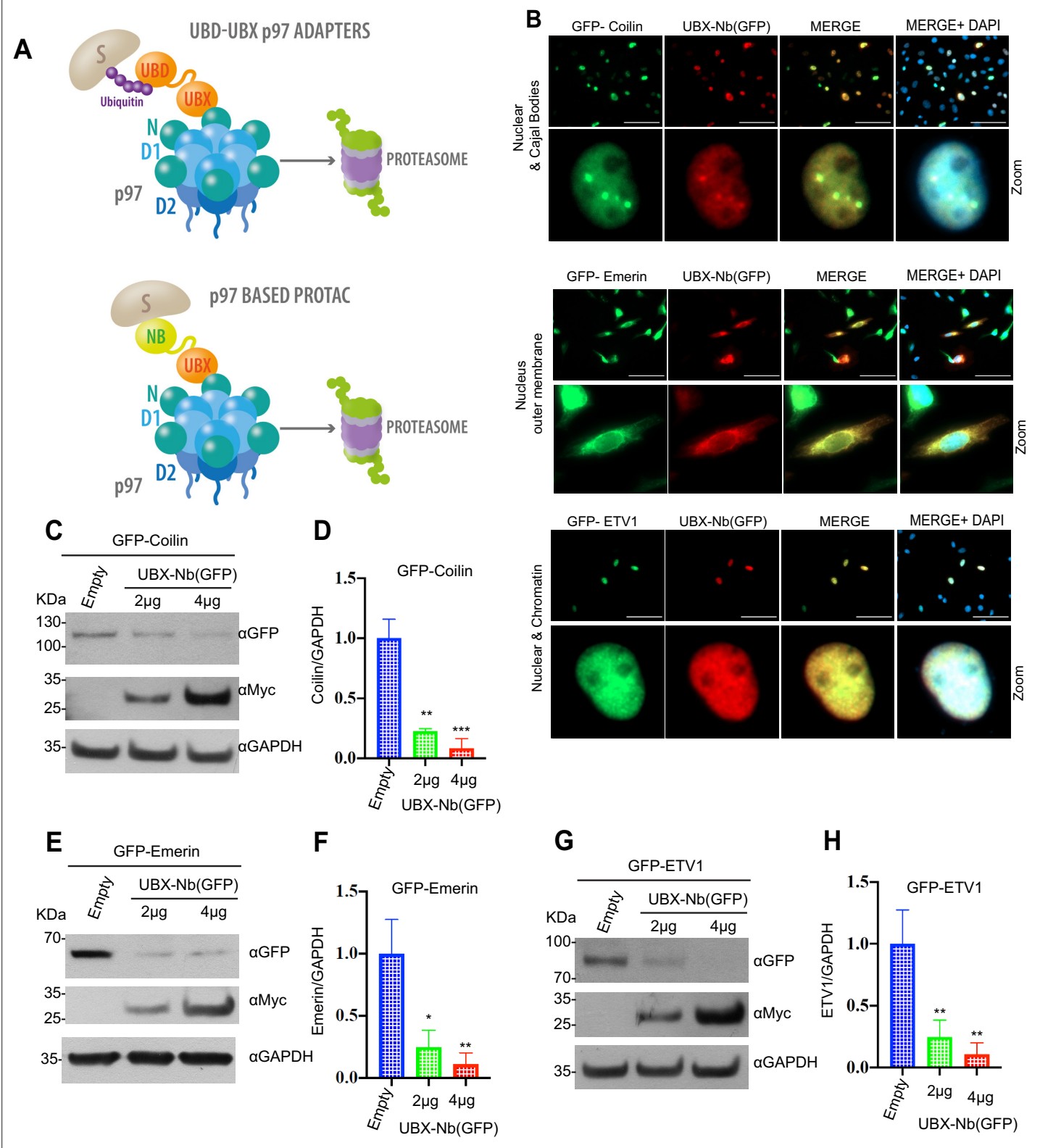

**Figure 1.** p97-mediated proteasome degradation. (**A**) Schematic representation of p97 presenting ubiquitinated proteins to the proteasome via a UBX domain-containing adaptor (top). The p97-PROTAC system, consisting of a UBX domain fused to a nanobody (Nb) that recruits substrates for p97-mediated segregation, unfolding, and proteasomal-mediated degradation (bottom). (**B**) p97-PROTAC (UBX-Nb(GFP)) recognizes GFP-tagged proteins at different cellular locations. HeLa cells were seeded on coverslips and co-transfected with UBX-Nb(GFP) and GFP-Coilin, GFP-Emerin, and GFP-ETV1. Cells

*Figure 1 continued*

were fixed and immunostained with anti-myc tag to verify the expression of UBX-Nb[(GFP)]. Colocalization was evaluated using ImageJ/Fiji with the Coloc 2 plugin, obtaining Pearson correlation coefficient (**R**) values as follows: GFP-Emerin=0.95, GFP-Coilin=0.96, and GFP-ETV1=0.95. (**C**) Western blot analysis of GFP-Coilin degradation by transfection with p97-PROTAC (UBX-Nb[(GFP)]). (**D**) Quantification of C. (**E**) Western blot analysis of GFP-Emerin degradation by transfection with p97-PROTAC (UBX-Nb[(GFP)]). (**F**) Quantification of E. (**G**) Western blot analysis of GFP-ETV1 degradation by transfection with p97-PROTAC (UBX-Nb[(GFP)]). (**H**) Quantification of G, *GFP-Coilin: 2 µg 'p-value' 0.0011 (\*\*), 4 µg 'p-value' 0.0009 (\*\*\*). GFP-Emerin: 2 µg 'p-value' 0.0130 (\*), 4 µg 'p-value' 0.0059 (\*\*). GFP-ETV1: 2 µg 'p-value' 0.0041 (\*\*), 4 µg 'p-value' 0.0020 (\*\*).* Western blots were quantified and statistically analyzed using a Student's t-test. p<0.05 compared to controls. n=3.

The online version of this article includes the following source data and figure supplement(s) for figure 1:

**Source data 1.** Raw Western blot images supporting GFP-Coilin, GFP-Emerin, and GFP-ETV1 degradation assays.

**Source data 2.** Annotated Western blot images indicating protein bands corresponding to GFP-Coilin, GFP-Emerin, and GFP-ETV1 degradation assays.

**Figure supplement 1.** p97-PROTAC sequence, colocalization analysis, and controls confirming UBX-dependent degradation.

**Figure supplement 1—source data 1.** Raw Western blots showing UBX-dependent degradation using an alternative nanobody as a control.

**Figure supplement 1—source data 2.** Annotated Western blots showing UBX-dependent degradation using an alternative nanobody as a control.

proteins within aggregates and packed in highly condensed regions, we chose 53BP1 as the target protein (encoded by the *TP53BP1* gene), as it is a well-known component of liquid-liquid phase separation (LLPS) structures (*Pessina et al., 2019*; *Zhang et al., 2022*).

First, we generated a knock-in (KI) cell line by fusing a YFP to the N-terminus of the endogenous 53BP1 protein using CRISPR/Cas9 technology. 24 hr after transfection with a forward and reverse gRNA and the donor vector, Cas9 D10A nickase was induced with doxycycline. Single YFP-positive cell clones were isolated by cell sorting (*Figure 2A*). Natural DNA damage that occurs during DNA replication induces the accumulation of 53BP1 fluorescence signal in distinct nuclear foci. The foci are typical for 53BP1, which has been described to accumulate in nuclear bodies during the G1 phase of the cell cycle (*Lukas et al., 2011*). The YFP-53BP1 KI clones were analyzed by fluorescence microscopy, and a heterozygote clone was selected (*Figure 2B*).

To further characterize 53BP1 foci in U2OS YFP-53BP1 (KI) cells, we employed structured illumination microscopy (SIM) (*Figure 2C*). We evaluated the localization of the YFP-53BP1 KI and endogenous 53BP1 by SIM and demonstrated the expected location for the KI fusion (*Figure 2D*). Accordingly, the YFP-53BP1 KI cell line fully recapitulated the localization of endogenous 53BP1, and the endogenous 53BP1 promoter controls the YFP-53BP1 KI expression.

53BP1 nuclear bodies have been described as membrane-less organelles, organized by LLPS. Hence, we used the YFP-53BP1 KI cells as a model to study the recruitment and function of the p97-PROTAC UBX-Nb[(GFP)] to target LLPS. We observed that the UBX-Nb[(GFP)] construct was successfully recruited to nuclear foci characterized by the 53BP1 location (*Figure 2E*). Importantly, UBX-Nb[(GFP)] expression significantly reduced 53BP1 levels, suggesting that our p97-PROTAC technology can specifically trigger the degradation of proteins within LLPS compartments (*Figure 2F and G*).

## Endogenous p97 expression in the brain

Conventional PROTAC technologies exploit the activity of E3 ubiquitin ligases and accordingly require the expression of a suitable ligase to cause target ubiquitination and degradation. To assess whether p97 could likewise be available for our approach, we studied the endogenous expression of p97 in the brains of animal models. Immunohistochemistry was performed using a rabbit polyclonal anti-p97 antibody (HPA012728, Sigma-Aldrich, whose specificity has been reported in the Human Protein Atlas). Consistent with these data, we observed ubiquitous expression in the cortex of nonhuman primates (NHPs), mice, and rats. Additionally, endogenous p97 expression was observed in the hippocampus and substantia nigra pars compacta (SNpc) of mice and rats, the major pathological sites in Alzheimer's and Parkinson's disease, respectively (*Figure 3*; *Gómez-Isla and Frosch, 2022*; *Blesa et al., 2022*). Unfortunately, no tissue samples of NHP were available for other brain regions. Thus, p97 is endogenously expressed in regions of clinical interest that are affected by neurodegenerative diseases. Ultimately, the *segregase* and *unfoldase* activity of p97 in the brain could be advantageous for p97-PROTAC-based targeted degradation of toxic protein aggregates, especially for neurodegenerative diseases.

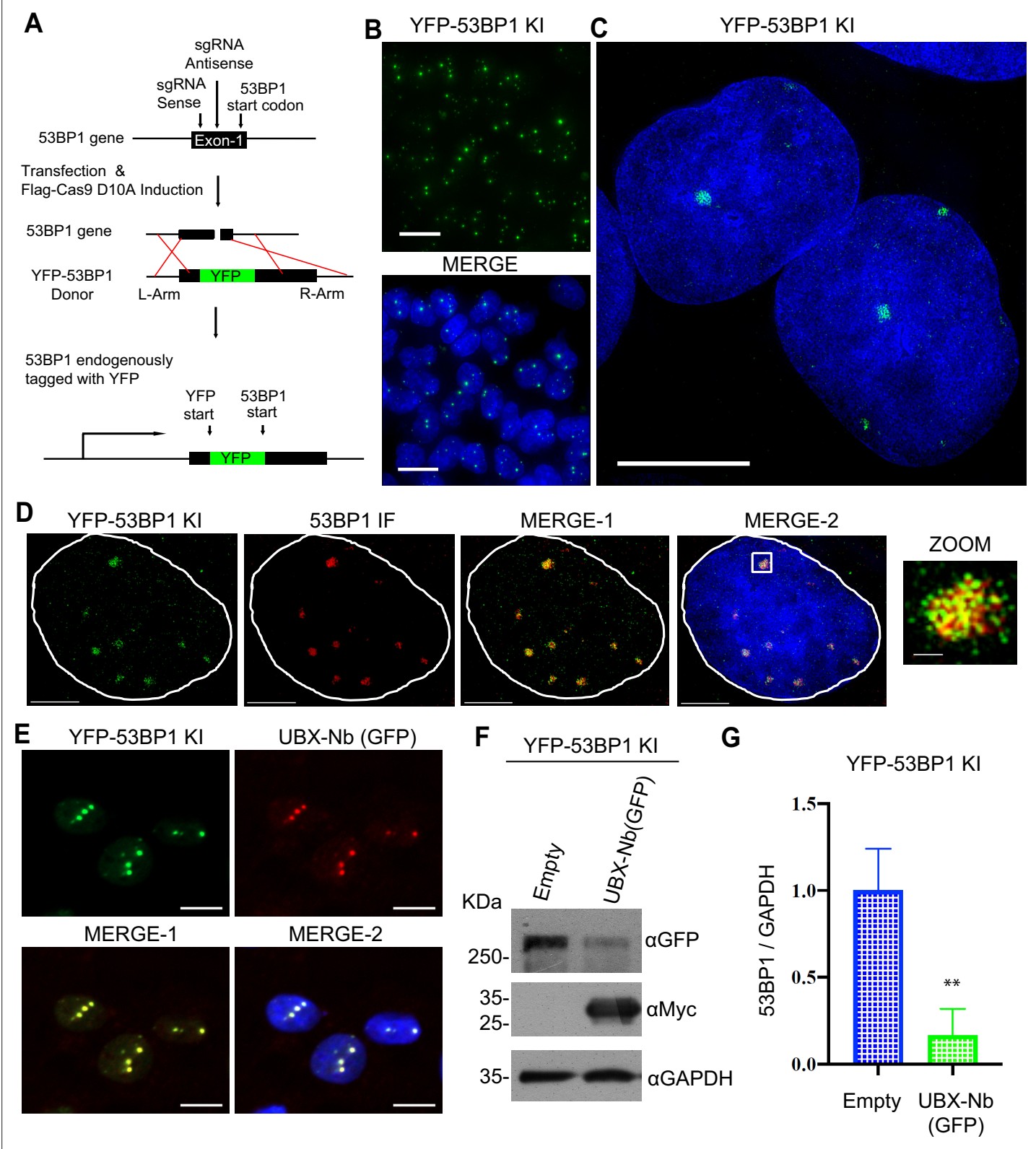

**Figure 2.** Targeting liquid-liquid phase separation proteins by a p97-PROTAC. (**A**) Strategy for inserting a yellow fluorescent protein (YFP) tag on the N-terminus of the 53BP1 gene in U2OS SEC-C cells using CRISPR/Cas9 D10A. (**B**) Selected knock-in (KI) YFP-53BP1 clones isolated via flow cytometry. Clones were confirmed via fluorescence microscopy (GE Deltavision Widefield). (**C**) Super-resolution images obtained with a Delta Vision OMX V4 structured illumination microscope (3D-SIM). (**D**) Immunofluorescence against 53BP1 (red) and colocalization with YFP-53BP1 in KI cells. Images were

*Figure 2 continued on next page*

*Figure 2 continued*

obtained using a Delta Vision OMX V4 structured illumination microscope (3D-SIM). (**E**) Recruitment of the p97-PROTAC UBX-Nb[(GFP)] (red) to YFP-53BP1 (green) within liquid-liquid phase separation structures. Data were obtained with a high-content CellDiscoverer 7. UBX-Nb[(GFP)] was detected using its myc-tag. (**F**) Western blot analysis of YFP-53BP1 degradation by UBX-Nb[(GFP)] transfection in the KI U2OS cells. (**G**) Quantification of F, '*p-value*' 0.0071 (**). Western blots were quantified and statistically analyzed using a Student's t-test. p<0.05 compared to controls. n=3.

The online version of this article includes the following source data for figure 2:

**Source data 1.** Raw Western blots showing p97-PROTAC–mediated degradation of YFP-53BP1 in KI U2OS cells.

**Source data 2.** Annotated Western blots indicating YFP-53BP1 degradation in KI U2OS cells transfected with p97-PROTAC.

## Structural and functional characterization of the p97-PROTAC mechanism

To gain insight into the structure of substrate-bound p97-PROTAC in the context of the p97 assembly, an AlphaFold2-based model of the p97-PROTAC/GFP-complex bound to p97 was generated (*Figure 4A*; see Materials and methods for details of the modeling procedure). After manually adjusting the conformation of the flexible linker between PROTAC UBX and Nb[(GFP)] moieties, the model shows that PROTAC-recruited GFP can access the central p97 pore (<15 Å distance between the pore entrance and the closest GFP residue) (*Figure 4A and B*). Consistent with the model, we determined that GFP alone, a barrel conformation protein of 28 kDa, can be recruited by the p97-PROTAC and is efficiently processed for degradation (*Figure 4C and D*).

To test the mechanism underlying by which the GFP became degraded, we co-transfected cells with GFP and either an empty vector or with the p97-PROTAC UBX-Nb[(GFP)]. After 20 hr, cells were treated with the proteasome inhibitor MG132 or DMSO as control. Upon MG132 treatment, we observed a full rescue of GFP protein levels. Therefore, by using a small substrate such as GFP as a model, we demonstrated that p97-PROTAC proteolytic activity relies on proteasome-mediated degradation (*Figure 4E and F*).

To determine whether p97 is directly involved in the degradation process mediated by the p97-PROTAC system, we compared cells co-transfected with the Myc-tagged nanobody anti-GFP (Nb-GFP-Myc) and a p97-GFP vector to cells co-transfected with Nb-GFP-Myc and an empty vector. Notably, cells expressing p97-GFP exhibited a significant reduction in Myc-tagged Nb-GFP levels. In contrast, cells transfected with the empty vector showed no effect on Myc-tag levels (*Figure 4—figure supplement 1A and B*). These results provide evidence supporting the direct role of p97 in facilitating the degradation mechanism mediated by the p97-PROTAC system.

To further validate these observations, we depleted p97 by transfecting cells with siRNAs targeting the *VCP* gene (*Figure 4G*). Upon p97 knockdown, a significant rescue of GFP-Emerin protein levels was observed, concomitant with an increase of the levels of the p97-PROTAC UBX-Nb[(GFP)] (*Figure 4H and I*), indicating that the PROTAC UBX-Nb[(GFP)] itself may also act as a substrate for p97.

To test the hypothesis that p97-PROTAC action is a ubiquitination-independent process, we globally inhibited ubiquitination using PYR41, a cell-permeable, irreversible inhibitor of the ubiquitin-activating enzyme E1 activity. First, we demonstrated that PYR41 was able to stabilize p53 in HeLa cells (*Figure 4—figure supplement 1C*). Next, we used GFP-Emerin as a substrate model and treated the cell with either DMSO or 50 µM of PYR41 for 4 hr. Consistent with our hypothesis, no inhibition of GFP-Emerin degradation was observed by the PYR41 treatment (*Figure 4J and K*). Finally, we investigated whether the activities of p97 were required for the substrate-mediated degradation triggered by p97-PROTAC UBX-Nb[(GFP)]. We tested the degradation of GFP-Emerin in HeLa cells treated with either DMSO or CB-5083, a potent inhibitor of the ATPase activity of p97. As an internal control, we measured the levels of CHOP, a sensor of ER stress known to be induced by p97 inhibitors *Figure 4L*. Surprisingly, we did not observe inhibition of degradation of GFP-Emerin in cells treated with CB-5083, despite the increase in CHOP levels (*Figure 4M*). Thus, our results show that p97-PROTAC targets proteins for proteasomal-mediated degradation by a ubiquitin-independent mechanism.

## Degradation of proteins of clinical interest by a p97-based PROTAC

A hallmark for several neurodegenerative diseases is the accumulation of toxic aggregates in the intra- or extracellular space (*Ross and Poirier, 2004*). PROTAC technology has the potential to become a new therapeutic approach for proteotoxic diseases (*Konstantinidou et al., 2019*). High levels of

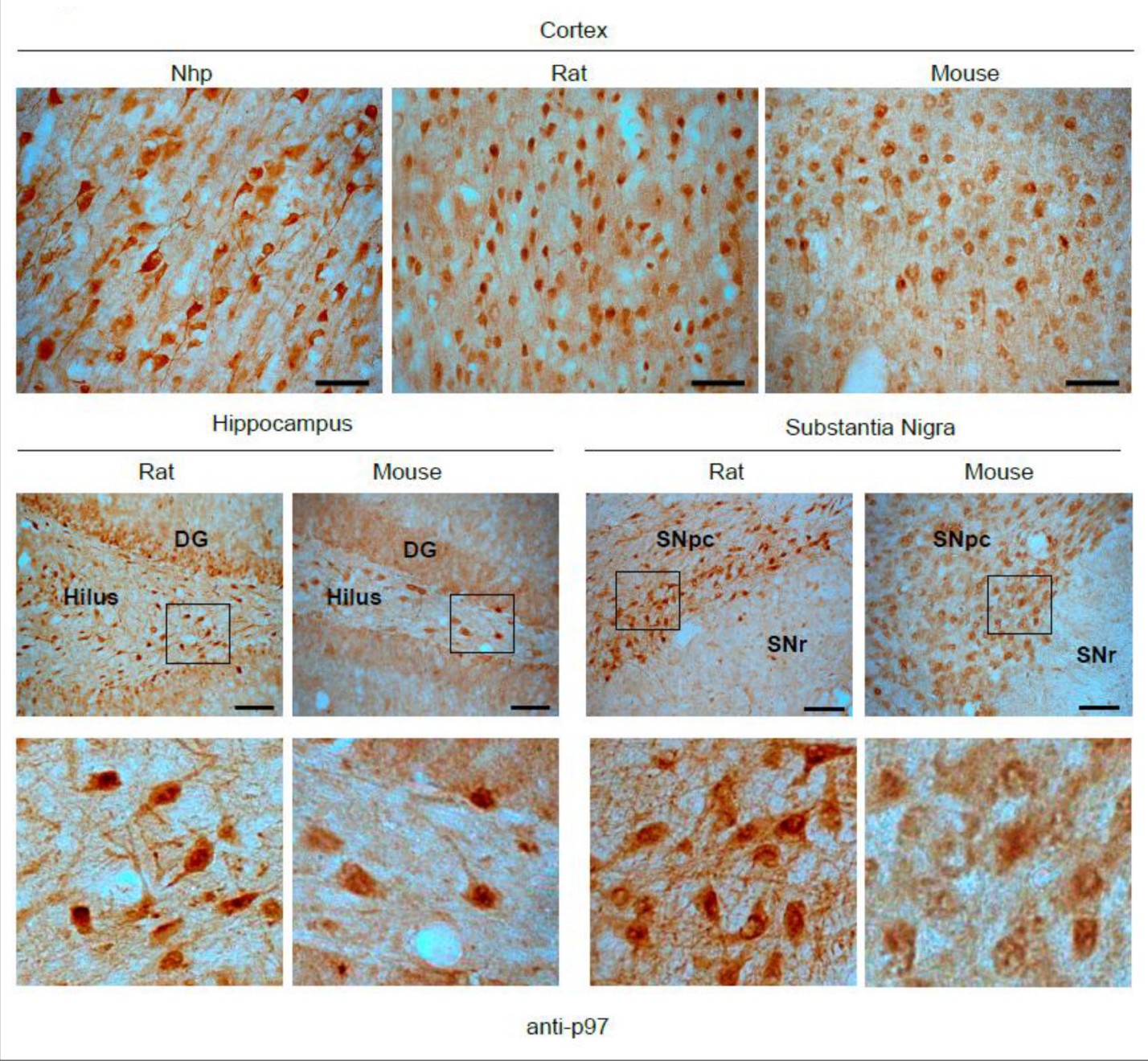

**Figure 3.** Endogenous p97 expression in brain tissue. Immunohistochemistry against p97 in brain tissue sections from nonhuman primates (Nhp) *Macaca fascicularis*, rat (*Sprague-Dawley*), and mouse (C57BL6/C). p97 expression was detected in substantia nigra pars compacta (SNpc), hippocampal, and cortical neurons. n=4.

ubiquitin are often found in intracellular aggregates, suggesting that ubiquitination-mediated proteasomal degradation may be partially impaired (*Galves et al., 2019*; *Gai et al., 2000*). Targeting p97 to toxic aggregates could potentially contribute to the clearance of aggregates by inducing segregation, unfolding, and proteasomal degradation simultaneously.

To evaluate the capacity of the p97-PROTAC system to degrade toxic aggregates, we tested two clinically relevant proteins: HTT and α-synuclein, which are known to form toxic aggregates. In the first experiment, we used GFP-fusion plasmids containing exon 1 of HTT with either 23 CAG repeats (EGFP-HTT Q23, wild-type HTT) or 74 CAG repeats (EGFP-HTT Q74, mutant HTT). We observed efficient degradation of both EGFP-HTT Q23 and EGFP-HTT Q74 by the p97-PROTAC UBX-Nb[(GFP)],

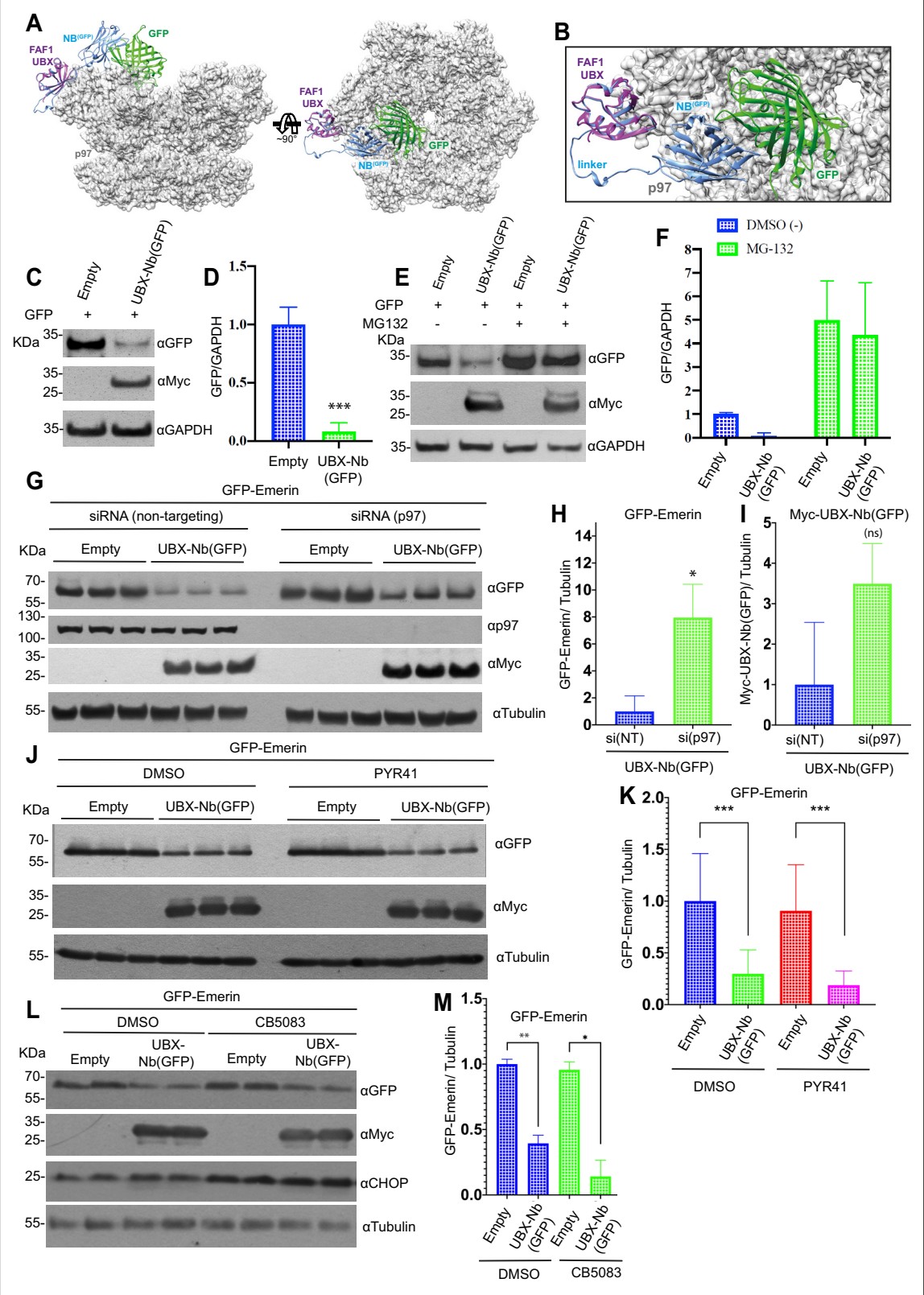

**Figure 4.** Molecular model and degradation activity of p97-PROTAC. (**A**) Model representations of the FAF1 UBX domain (purple), UBX-Nb[(GFP)] (blue), GFP (green), and the p97 hexamer (light gray cartoon with semitransparent molecular surface representation). (**B**) A magnified view of the model shown in A. (**C**) GFP monomer was co-transfected with UBX-Nb [(GFP)] or empty vector in HeLa cells. Protein degradation was analyzed by western blot analysis. (**D**) Quantification of C, 'p-value' 0.0007 (***). (**E**) GFP monomer was co-transfected with UBX-Nb[(GFP)] or empty vector in HeLa cells, after 24 hr, the cells

*Figure 4 continued on next page*

*Figure 4 continued*

were incubated with DMSO or the proteasome inhibitor MG132 (25 µM final concentration) for 4 hr. Protein degradation was analyzed by western blot. (**F**) Quantification of E, '*p-value*' DMSO treatment 0.0120 (*), '*p-value*' MG132 treatment 0.7776 (ns). (**G**) p97 was silenced by transfection with *VCP*-siRNA in HeLa cells. Subsequently, the cells were transfected with GFP-Emerin and either the empty vector or UBX-Nb[(GFP)] vector. Protein degradation was analyzed by western blot analysis. (**H**) Quantification of GFP-Emerin in UBX-Nb[(GFP)] cells treated with either siNT control or a *VCP*-siRNA, '*p-value*' 0.0114 (*). (**I**) Quantification of the UBX-Nb[(GFP)] (myc-tag) in UBX-Nb[(GFP)] cells treated with either siNT control or a *VCP*-siRNA, '*p-value*' 0.0775 (ns). (**J**) HeLa cells co-transfected with GFP-Emerin and either empty vector or UBX-Nb[(GFP)]; in addition, the cells were treated with the E1 ubiquitin inhibitor PYR-41 (50 µM) for 4 hr at 37°C. Subsequently, total proteins were extracted, and protein degradation was analyzed by western blot. (**K**) Quantification of J, '*p-value*' DMSO treatment 0.0009 (***), '*p-value*' PYR-41 treatment 0.0003 (***). (**L**) GFP-Emerin was co-transfected with UBX-Nb[(GFP)] or empty vector in HeLa cells, after 24 hr the cells were incubated with DMSO (as control) or the p97 inhibitor CB-5083 (4 µM final concentration) for 6 hr. Protein degradation was analyzed by western blot using total proteins. (**M**) Quantification of L, '*p-value*' DMSO treatment 0.0071 (**), '*p-value*' CB-5083 treatment 0.0139 (*). Western blots were quantified and statistically analyzed using a Student's t-test. p<0.05 compared to controls. n=3.

The online version of this article includes the following source data and figure supplement(s) for figure 4:

**Source data 1.** Raw Western blots showing p97-PROTAC–mediated degradation using inhibitors and siRNA controls.

**Source data 2.** Annotated Western blots showing p97-PROTAC–mediated degradation under inhibitor and siRNA conditions.

**Figure supplement 1.** Validation of p97-dependent degradation and evaluation of PYR41 treatment.

**Figure supplement 1—source data 1.** Raw Western blots showing p97-dependent degradation and PYR41 treatment effects.

**Figure supplement 1—source data 2.** Annotated Western blots indicating protein bands for p97-dependent degradation and PYR41 treatment effects.

as demonstrated in *Figure 5D and E*, respectively. Also, in both cases, co-localization of the p97-PROTAC with EGFP-HTT was evident within the cells (*Figure 5C and F*). Notably, EGFP-HTT Q74 transfection resulted in the formation of aggregates, and the p97-PROTAC was observed to co-localize with these aggregates, supporting the direct interaction of the p97-PROTAC with the target protein, regardless of its aggregation state.

Since Huntingtin aggregates were undetectable by western blot assays using these vectors, we utilized an EGFP-fusion plasmid containing the exon 1 of HTT with 24 CAG repeats. HeLa cells were co-transfected with EGFP-Q24 and either the p97-PROTAC UBX-Nb[(GFP)] or an empty vector. The UBX system efficiently degraded Q24, as evidenced by a significant reduction in the levels of both monomeric protein and aggregates. Upon overexposure of the film, high-molecular-weight EGFP-Q24 species were detected. These bands were notably attenuated in cells treated with the UBX-Nb[(GFP)], demonstrating that the p97-PROTAC effectively reduces the expression of Q24 and its aggregates (*Figure 5—figure supplement 1*), reinforcing the observations obtained by EGFP-Q23 and Q74.

To validate the versatility of the p97-PROTAC system, we generated an anti-α-synuclein p97-PROTAC by replacing the anti-GFP nanobody with a specific α-synuclein nanobody (NbSyn87) (*Butler et al., 2016*; *El Turk et al., 2018*; *Chatterjee et al., 2018*; *Guilliams et al., 2013*). We tested the system using both GFP-tagged and untagged α-synuclein A53T mutant by co-transfecting cells with increasing concentrations of the p97-PROTAC UBX-Nb[(Syn87)] or an empty vector. In both cases, we observed efficient degradation of the α-synuclein A53T (*Figure 6A–D*). As an additional measure to detect and quantify α-synuclein aggregation in our system, we performed aggregation assays using Thioflavin T (ThT), which is a small molecule that can bind specifically to β-sheet-rich structures of amyloid fibrils, enabling fluorescence-based detection. As shown in *Figure 6E*, aggregates were successfully detected in cells transfected with wild-type α-synuclein, and notably, the levels of these aggregates were reduced in cells that were co-transfected with p97-PROTAC UBX-Nb[(Syn87)] or with p97-PROTAC UBX-Nb[(GFP)]. Thus, we demonstrated that the nanobody component of the p97-PROTAC system is exchangeable, and the p97-PROTAC system is suitable for degrading human proteins of clinical interest.

## Discussion

The constitutive degradation of proteins is a rapid negative-regulatory mechanism, often involved in important stress response pathways. For example, under normoxia, Hif1α is rapidly hydroxylated in an oxygen-dependent manner, and the hydroxylation is recognized by an E3 ligase receptor called VHL leading to continuous ubiquitin-mediated proteasomal degradation (*Min et al., 2002*; *McGettrick and O'Neill, 2020*; *Masoud and Li, 2015*; *Yeo, 2019*). Hypoxia decreases Hif1α hydroxylation, and

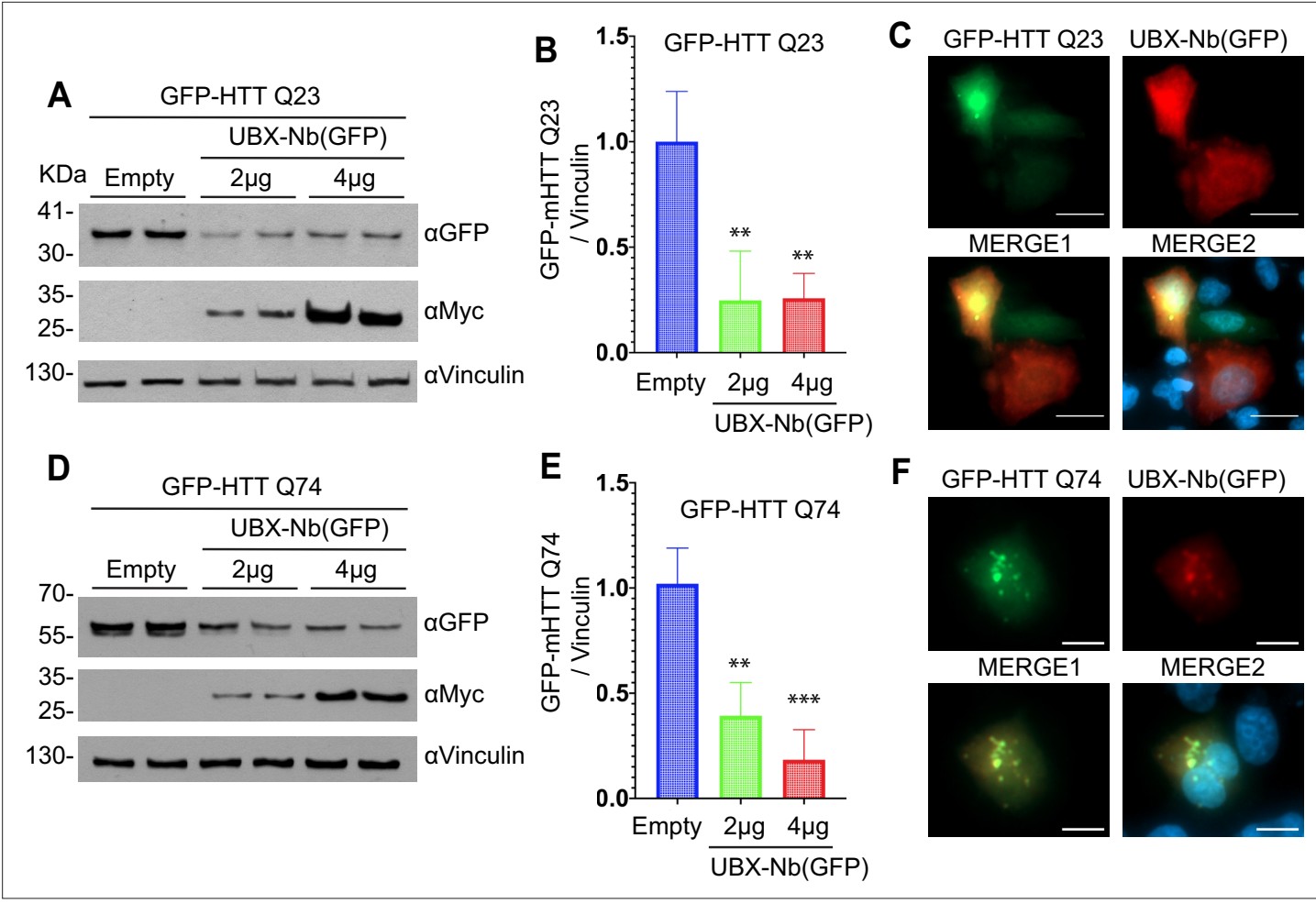

**Figure 5.** Degradation of Huntingtin wild type and mutant with p97-PROTAC UBX-Nb(GFP). HeLa cells were transiently co-transfected with HTT GFP-tagged plasmids containing either 23 CAG repeats (EGFP-HTT^Q23-wild-type HTT), 74 CAG repeats (EGFP-HTT^Q74: mutant HTT), or 24 CAG repeats (EGFP-HTT^Q24). (**A**) Cells were co-transfected with GFP-HTT^Q23 and increasing amounts of the p97 PROTAC UBX-Nb(GFP), degradation was determined by western blot analysis. (**B**) Quantification of A, *2 µg 'p-value' 0.0041 (**), 4 µg 'p-value' 0.0014 (**)*. (**C**) Immunofluorescence showing the recruitment of p97 PROTAC UBX-Nb(GFP) to GFP-HTT^Q23. (**D**) HeLa cells were co-transfected with GFP-HTT^Q74 and increasing amounts of the p97 PROTAC UBX-Nb(GFP), degradation was determined by western blot analysis. (**E**) Quantification of D, *2 µg 'p-value' 0.0016 (**), 4 µg 'p-value' 0.0003 (***)*. (**F**) Immunofluorescence to demonstrate the recruitment of p97 PROTAC UBX-Nb(GFP) to GFP-HTT^Q74. Western blots were quantified and statistically analyzed using a Student's t-test. $p < 0.05$ compared to controls. n=3.

The online version of this article includes the following source data and figure supplement(s) for figure 5:

**Source data 1.** Raw Western blots showing p97-PROTAC–mediated degradation of GFP-HTTQ23 and GFP-HTTQ74.

**Source data 2.** Annotated Western blots indicating protein bands for p97-PROTAC–mediated degradation of GFP-HTTQ23 and GFP-HTTQ74.

**Figure supplement 1.** HeLa cells were co-transfected with GFP-HTT Q24 and either the p97-PROTAC UBX-Nb(GFP) or an empty vector.

**Figure supplement 1—source data 1.** Raw Western blots showing p97-PROTAC effects on GFP-HTTQ24 degradation and aggregate levels.

**Figure supplement 1—source data 2.** Annotated Western blots indicating protein bands for GFP-HTTQ24 degradation and aggregate analysis under p97-PROTAC treatment.

consequently, Hif1α dissociates from VHL, resulting in rapid stabilization, accumulation, and activation of the hypoxia response. Therefore, endogenous degradation is a natural and efficient way to down-regulate the amount and function of proteins.

PROTAC technologies today rely on ubiquitin-mediated proteasomal degradation, but abnormal protein aggregation has multiple detrimental effects in the ubiquitin-mediated proteasomal degradation, including a significant reduction in the free ubiquitin pool and abundant ubiquitin accumulation in proteotoxic aggregates (*Ben Yehuda et al., 2017*). Our p97-PROTAC technology provides a

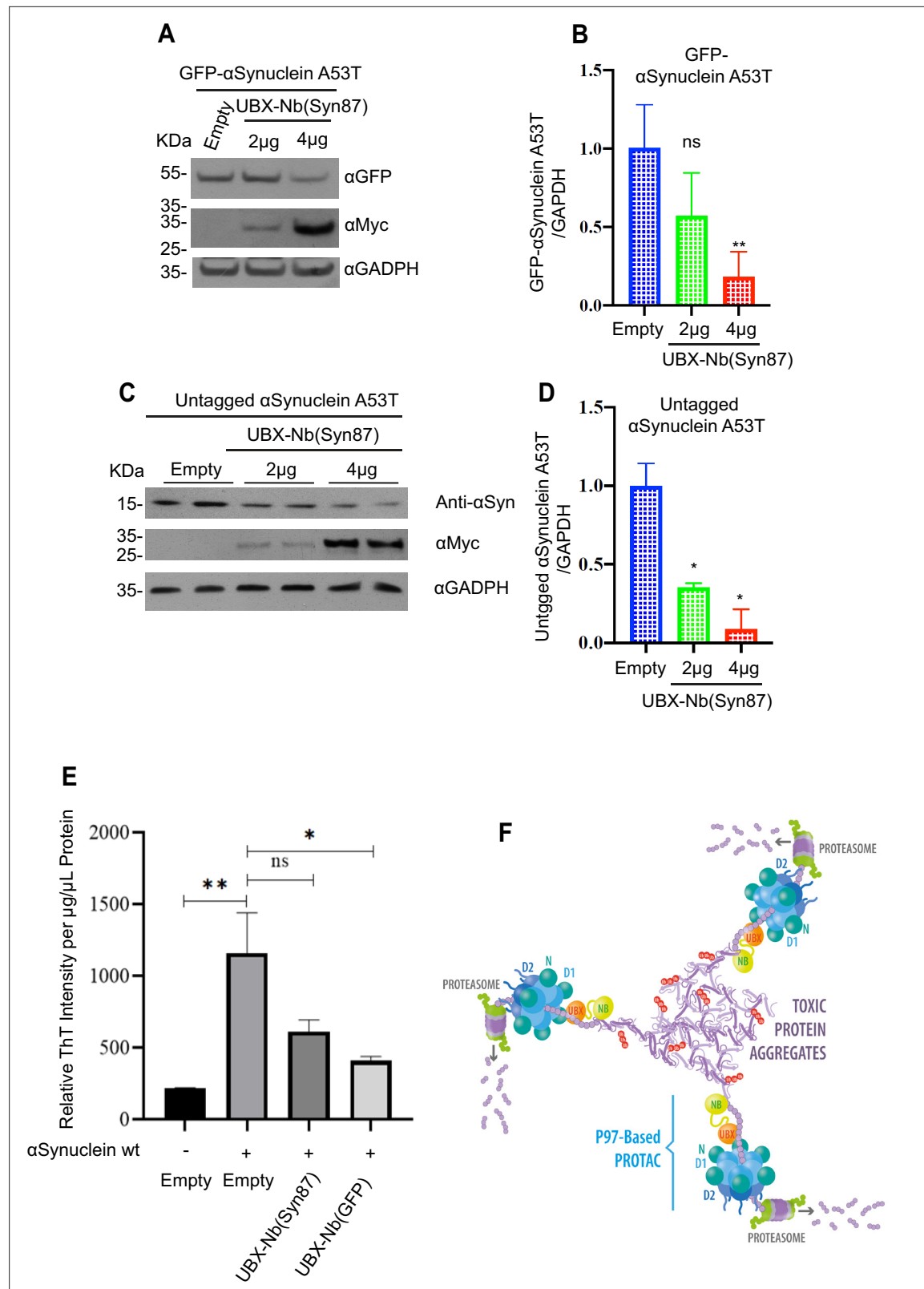

**Figure 6.** Degradation of α-synuclein with p97-PROTAC. (**A**) HeLa cells were co-transfected with a vector expressing αSynuclein mutant A53T fused to GFP (GFP-αSynuclein A53T) and empty or increasing concentrations of UBX-Nb[(Syn87)]. A53T-GFP degradation was determined by western blot. (**B**) Quantification of A, *2 µg 'p-value' 0.0651 (ns), 4 µg 'p-value' 0.0020 (**).* (**C**) Cells were co-transfected with a vector expressing untagged α-synuclein mutant A53T and an empty vector or increasing concentrations of UBX-Nb[(Syn87)]. Untagged αSynuclein A53T degradation was determined by western

*Figure 6 continued on next page*

*Figure 6 continued*

blot using an anti αSynuclein antibody. (**D**) Quantification of C, *2 µg 'p-value' 0.0240 (\*), 4 µg 'p-value' 0.0210 (\*)*. (**E**) Quantification of α-synuclein aggregation via Thioflavin T (ThT) fluorescence. Non-transfected cells and cells transfected with GFP-tagged wild-type α-synuclein (pcDNA5 WT αSyn-GFP), either alone or co-transfected with the p97-based PROTACs containing nanobodies against GFP [UBX-Nb[(GFP)]] or α-synuclein [UBX-Nb[(Syn87)]], were analyzed for relative ThT fluorescence intensity normalized to total protein concentration (µg/µL). Bars represent mean ± SEM. Asterisks (\*) indicate statistically significant differences compared to control (\*p<0.05, Dunnett's post hoc test); 'ns' indicates no significant difference. (**F**) Representative model of p97-PROTAC functioning in the degradation of proteins and protein aggregates. Western blots were quantified and statistically analyzed using a Student's t-test. p<0.05 compared to controls. n=3.

The online version of this article includes the following source data for figure 6:

**Source data 1.** Raw Western blots showing p97-PROTAC–mediated degradation of GFP-tagged and untagged α-synuclein A53T.

**Source data 2.** Annotated Western blots indicating protein bands for p97-PROTAC–mediated degradation of GFP-tagged and untagged α-synuclein A53T.

promising alternative to ubiquitin-mediated degradation, offering a novel approach to overcoming these challenges.

In this study, we demonstrated that an engineered p97-PROTAC consisting of a UBX domain fused to a camelid nanobody effectively induces specific protein degradation in cells. Using siRNA specific against the *VCP* gene, we abrogated substrate degradation, confirming the involvement of p97 in the process. Complementary to the siRNA assay, we utilized a p97-GFP vector, which further demonstrated that p97 contributes to the degradation of target proteins. These findings reinforce the central role of p97 in the functionality of the p97-PROTAC system.

In addition, we demonstrated that the degradation mechanism operates independently of ubiquitination, as shown by utilizing the ubiquitin E1 inhibitor PYR41. It is well established that different pools of the same protein can be directed to the proteasome via both ubiquitin-dependent and ubiquitin-independent mechanisms under the same cellular conditions (*Ju and Xie, 2004*). Our findings are consistent with prior studies showing that various proteins can be degraded by the proteasome without ubiquitin tagging. Notably, intrinsically disordered regions or transiently unfolded states in proteins facilitate their recognition and degradation by the proteasome in a ubiquitin-independent manner (*Butler et al., 2016*; *Erales and Coffino, 2014*; *Makaros et al., 2023*; *Li et al., 2025*; *Bialek et al., 2023*). In this context, p97's ability to unfold substrates may play a critical role in exposing these unstructured regions, thereby enabling their recognition and processing by the proteasome.

Intriguingly, we observed that the activity of p97-PROTACs was not inhibited by the specific p97 ATPase inhibitor CB-5083. Thus, mechanistically, we cannot distinguish if the p97-PROTACs overcame the effect of the inhibitor or if the degradation mediated by the p97-PROTACs is mediated only by induced proximity to the proteasome and independent of p97's ATPase activity. Although p97 has classically been described as a *segregase* that drives proteasomal degradation, it also plays critical roles in lysosomal and autophagic pathways (*Meyer et al., 2012*; *Körner et al., 2025*). This raises the possibility that substrate recruitment to p97 via our PROTAC system may activate additional degradation pathways. The observation that inhibition of the D2 domain by CB-5083 did not abolish degradation is consistent with this view and suggests either incomplete inhibition under our conditions or the involvement of other functions of p97, such as scaffolding properties or contributions from the D1 domain. Supporting this broader perspective, a 2021 study showed that gossypol, a clinically approved drug in China, reduces mHTT levels and improves motor function in Huntington's disease models by stabilizing a VCP-LC3-mHTT ternary complex, thereby promoting autophagic clearance of mHTT (*Li et al., 2021*). Moreover, ATP binding to the D1 domain influences adaptor selection, suggesting that D1 regulates cofactor engagement and may contribute to proteostasis through noncanonical mechanisms (*Chia et al., 2012*). Together with recent structural insights into inter-domain communication within p97 (*Shein et al., 2024*; *Turner et al., 2025*), these findings support a model in which UBX adaptors can exploit both p97-dependent and partially independent mechanisms in a context-dependent manner. In our engineered PROTAC system, UBX recruitment enables degradation even under conditions where D2 activity is inhibited, consistent with a role for D1- or scaffold-mediated functions of p97. We therefore propose that UBX domains constitute a versatile platform capable of linking substrates either to canonical p97 *unfoldase* activity or to alternative degradation routes. Taken together, these results underscore the multifunctional nature of p97 and suggest that our technology may exploit not only the classical proteasomal pathway but also additional p97-dependent

clearance mechanisms. However, further studies will be required to define the precise mechanisms by which p97-PROTACs mediate ubiquitin-independent degradation in cells.

p97 is an extensively characterized ubiquitously expressed protein involved in fundamental cellular processes, such as the degradation of proteins associated with the endoplasmic reticulum (ER) (ERAD) (*Hyun and Shin, 2021*), autophagy (*Creekmore et al., 2024*), and it is also an important player in the proteostasis of aggregates in Parkinson's disease (*Alieva et al., 2020*; *Merchant et al., 2019*; *Lang and Espay, 2018*). Indeed, p97 has been shown to protect against the proteopathic spread of pathogenic aggregates in animal models (*Zhu et al., 2021*). In this study, we characterized p97 expression in the brains of animal models and observed ubiquitous p97 expression in all sections. Specifically, high p97 expression levels were detected in the cortex, SNpc, and hippocampus, regions highly prone to protein aggregation and neurodegeneration in neurodegenerative diseases such as Parkinson's and Alzheimer's disease (*Ross and Poirier, 2004*). These observations support the possibility of using p97-based PROTACs to reduce protein aggregation in neurodegeneration-related targets in the future.

The PROTAC technologies might increase the number of druggable proteins by a change in the inhibition paradigm. In contrast to conventional drugs that aim for chemical modulation of the enzymatic activity or function, PROTAC aims at the downregulation of the levels of specific targets based exclusively on selective binding and degradation.

In conclusion, our work unveils the potential of a p97-PROTAC as the first E3 ubiquitin ligase-independent technology that targets proteins for degradation at diverse subcellular locations, including integral membrane protein residing at the inner nuclear membrane, chromatin-located, and liquid-liquid phase-separated compartments. Also, it provides a new technology to target protein aggregates of clinical interest for proteasomal degradation. These findings establish p97-PROTAC as a versatile and promising platform for future therapeutic applications.

## Materials and methods
### Design and cloning
The constructs were designed to include three main domains: an anti-GFP nanobody (*Fulcher et al., 2016*), which specifically binds to the target protein; a linker and the UBX domain of the p97 adaptor FAF1. The linker used to connect both domains (KESGSVSSEQLAQFRSLD) was originally designed for the construction of single-chain antigen-binding proteins, providing sufficient flexibility to connect the domains without hindering the antigen-binding capacity (*Bird et al., 1988*). The anti-GFP nanobody is capable of recognizing both YFP and GFP proteins. Additionally, a myc-tag was incorporated to confirm expression. The full sequence was cloned into the pcDNA 5 FRT/TO vector by the Gibson Assembly technique. Later, the anti-GFP nanobody was replaced with an anti-α-synuclein nanobody (NbSyn87) using restriction enzymes (*Butler et al., 2016*; *El Turk et al., 2018*; *Chatterjee et al., 2018*; *Guilliams et al., 2013*). Furthermore, the UBX domain was removed by restriction enzyme digestion, resulting in the Nb-anti-GFP-Myc-tag vector.

### Cell culture
HeLa and U2OS cells were generously provided by Professor Ronald T. Hay (University of Dundee). Authentication certificates were not available; however, the morphology and growth characteristics of both lines were consistent with those expected for HeLa and U2OS cells. Cultures were routinely screened and tested negative for mycoplasma contamination. Cells were cultured in Dulbecco's modified Eagle's medium (DMEM; Gibco) supplemented with 10% fetal bovine serum and 100 units/mL penicillin–streptomycin and maintained at 37°C in a humidified incubator with 5% $CO_2$. Plasmid transfection was performed in six-well plates using 4 µg of DNA, 24 hr after transfection, cells were lysed, and proteins were collected. Cells were transiently transfected with the following vectors: pcDNA5 FRT/TO GFP-ETV1, pcDNA5 FRT/TO GFP-Emerin, pEYFP-Coilin, pcDNA FRT/TO GFP, VCP(wt)-EGFP (Addgene, #23971), pEGFP-C1-tagged plasmids containing the exon 1 of HTT with 23 CAG repeats (pEGFP-Q23: wild-type HTT, Addgene, #40261) or 74 CAG repeats (pEGFP-Q74: mutant HTT, Addgene, #40262) and pEGFP-Q24 (generously provided by Dr. Maite Castro, Universidad Austral de Chile). Vectors to α-synuclein were a gift from our collaborator Dr. Gopal Sapkota (pcDNA5-FRT/TO-GFP-SCNA-A53T, #59047 or pcDNA5-FRT/TO-SCNA-A53T, #59042). As a control, we used an empty vector pcDNA5 FRT/TO. Transfection was performed using Lipofectamine 2000 (Invitrogen)

according to the manufacturer's instructions, and media were supplemented with normocin (100 µg/mL) during transfection (Invivogen). For lysis, cells were washed twice in ice-cold phosphate-buffered saline (PBS), scraped on ice in lysis buffer (Tris-HCl pH 6.8, NaCl, glycerol, and SDS 10%), supplemented with complete protease inhibitors (one tablet per 25 mL: Roche), and 0.1% β-mercaptoethanol (Sigma). Cell extracts were either cleared and processed immediately or stored at –20°C. The protein concentration was determined in a 96-well format using the Pierce BCA protein assay kit (Thermo Fisher Scientific).

## Antibodies and inhibitors

We used the following primary antibodies: anti-GFP (Invitrogen, GF28R mAb MA5-15256), anti-α-synuclein (Santa Cruz Biotechnology, 3H2897 mAb sc-69977), anti-53BP1 (Invitrogen, 53BP1 Polyclonal Antibody PA1-16565), anti-Myc-tag (Cell Signaling, 9B11 Mouse mAb #2276), anti-vcp/p97 (Sigma-Aldrich Polyclonal Antibody HPA012728), anti-p53 (Invitrogen, monoclonal antibody DO-7 #MA5-12557), anti-CHOP (Cell Signaling, L63F7 Mouse mAb #2895), anti-ubiquitin (Sigma-Aldrich ST1200 Mouse mAb FK2), anti-α-tubulin (Santa Cruz Biotechnology, B7 mAb sc-5286), anti-GAPDH (Santa Cruz Biotechnology, mAb sc-47724), anti-vinculin (Santa Cruz Biotechnology, 7F9 mAb sc-73614). Horseradish peroxidase (HRP)-coupled secondary antibodies and Alexa Fluor secondary antibodies were purchased from Invitrogen (Thermo Fisher): Mouse IgG, IgM (H+L) Secondary Antibody (A-10677), Rabbit IgG (H+L) Secondary Antibody (31460), and Mouse IgG (H+L) Highly Cross-Adsorbed Secondary Antibody (A-11032).

The inhibitors used in this work were: MG132, a proteasome inhibitor from Sigma-Aldrich (catalog number 474790); PYR-41, an E1 ligase inhibitor from Sigma-Aldrich (catalog number N2915); and CB-5085, a D2 domain inhibitor of p97 from Cayman Chemical.

## Generation of a 53BP1 endogenously tagged YFP-53BP1 using CRISPR/Cas9

U2OS T-Rex cells were co-transfected with the pOG44 plasmid, which constitutively expresses the Flp recombinase and pcDNA5 FRT/TO codon optimized *Streptococcus pyogenes* M1 GAS Cas9 D10A-NLS-FLAG (DU45732 MRC-PPU reagent) for isogenic integration of the cassette. To decrease possible off-target effects, we applied the Cas9 D10A nickase system for genetic cleavage. Single clones were selected to generate the U2OS stable expressing Cas9 D10A cells, U2OS SEC-C D10A. Two specific guide RNAs (gRNAs) targeting sequences were identified in the 5' UTR before the 53BP1 start codon using the E-CRISPR software (http://www.e-crisp.org/E-CRISP/reannotate_crispr.html). The forward gRNA g53BP1-1 5'AGACCTCTAGCTCGAGCGCGAGG 3' and a reverse gRNA g53BP1-7 5'GTCCCTCCAGATCGATCCCTAGG 3' were cloned into pU6-Puro via site-directed mutagenesis using the QuickChange method (Stratagene) cloned using the previously described methodology and confirmed by DNA sequencing (*Rojas-Fernandez et al., 2015*; *Munoz et al., 2014*). We designed a strategy for YFP delivery at the N-terminus of the 53BP1 protein using a CRISPR/Cas9 KI methodology. In short, the sense and antisense gRNAs were transfected in U2OS T-Rex cells and modified to produce Cas9 D10A nickase in a doxycycline-inducible manner. We also engineered a synthetic vector with two homologous flanking regions around the cleavage site, where we inserted YFP cDNA and small linker in front of the endogenous 53BP1 gene synthetically produced by GeneArt (Life Technologies). The gRNA recognition sites were mutated on the synthetic donor to avoid cleavage by the gRNA/Cas9 complex. Finally, the two gRNAs and the donor vector carrying the YFP cDNA flanked with the homologue regions were transfected in U2OS cells. The day after the cells were transfected, the expression of the Cas9 nuclease was induced by adding 1 µg/mL doxycycline, and the cells were incubated for 4 days before FACS analysis to identify YFP-positive cells and further cell sorting based on single-cell isolation.

## Flow cytometry and cell sorting

Cells were analyzed for YFP fluorescence on an LSR Fortessa or FACS Canto flow cytometer (Becton Dickinson), and data were analyzed using FlowJo software (Tree Star Inc). Single cells were identified based on FSC-A, FSC-W, and SSC-A, and YFP fluorescence measured with 488 nm excitation and emission detected at 530±30 nm. Cell sorting was performed on an Influx cell sorter (Becton Dickinson) with FACS software, using the same cell identification procedure described above. Single

YFP-expressing cells were sorted onto individual wells of a 96-well plate containing DMEM supplemented with 20% fetal bovine serum (FBS), 2 mM L-glutamine, 100 units/mL penicillin, 100 µg/mL streptomycin, and 100 µg/mL transfection-compatible antibiotic normocin 1× (Invivogene). Single-cell clones were left to proliferate, and YFP fluorescence was determined.

## Immunofluorescence and high-content microscopy

HeLa cells transfected with GFP proteins were grown in a 96-well optical plate (Thermo Fisher Scientific) or 24-well lidded plates. Cells were fixed with 4% paraformaldehyde at 37°C for 10 min. Cells were washed with 1× PBS and permeabilized in 0.2% Triton X-100 PBS. After washing the cells three times in 1× PBS, they were incubated with blocking solution (FBS 5%-PBS 1×) for 30 min and then incubated with the primary antibody for 1 hr at 37°C. After washing another three times with 1× PBS, a secondary antibody Alexa Fluor was used at 1: 3000, it was incubated for 45 min at 37°C. For nuclei staining, cells were washed with PBS and incubated for 10 min at room temperature with 0.1 mg/mL DAPI. After the final wash, the cells were kept in the 96-well optical plates in 1× PBS. Cover-fixed cells were mounted on slides. Fixed cell images were acquired with a high-content automated microscope, CellDiscoverer 7 (Carl Zeiss GmbH, Jena, Germany).

## Silencing assay

HeLa cells were cultured in a six-well plate and transfected with siRNAs control siRNA-B, (sc-44230), or *VCP* siRNA (sc-37187) at 40% confluence. Transfection was performed using Lipofectamine RNAiMax (Invitrogen) according to the manufacturer's instructions, and the media were supplemented with Normocin (Invivogen) during transfection. Twenty-four hours post-siRNA transfection, the medium was changed to fresh DMEM-Normocin, and GFP-Emerin, as well as both the empty vector and the UBX-Nb$^{(GFP)}$ vector, was co-transfected. The complexes were left in culture for 48 hr. Subsequently, total proteins were extracted with lysis buffer to perform the western blot assay.

## ThT fluorescence measurement protocol

Fluorescence measurements were performed using a FluoroStar Omega microplate reader (BMG Labtech, Germany) and a 96-well SPL Black plate (SPL Life Sciences, Korea). Each reaction well contained a final volume of 100 µL, consisting of 15 µL of cell culture, Thioflavin T (ThT, Sigma-Aldrich) at a final concentration of 15 µM, and PBS pH 7 (Gibco, Thermo Fisher Scientific). Prior to measurement, samples were homogenized by double orbital shaking at 300 rpm for 60 s. The fluorescence intensity mode was used, with spiral averaging and a scan diameter of 10 mm. A single read cycle was performed with 100 flashes per well to enhance measurement accuracy. Optical settings were configured with an excitation wavelength of 448±10 nm and an emission wavelength of 482±10 nm. Fluorescence values were normalized to total protein concentration, determined using the BCA assay (Biovision, USA), and relative ThT fluorescence was expressed as a function of protein concentration (µg/µL).

## SDS-PAGE and western blotting

Reduced protein extracts were separated on 10% SDS-PAGE gels or 4–12% NuPAGE Bis–Tris precast gels (Invitrogen) by electrophoresis. Proteins were then transferred onto nitrocellulose membranes (Millipore). Nonspecific sites were blocked with blocking solution (PBS containing 0.1% Tween 20 with 5% [wt/vol] nonfat milk) for 30 min at room temperature with agitation. The blocking solution was discarded, and membranes were incubated overnight at 4°C in 5% BSA PBS-T (PBS 1× containing 0.1% Tween 20) with the appropriate primary antibodies. Membranes were then washed in PBS-T and incubated with HRP-conjugated secondary antibodies in 5% milk-PBS-T for 1 hr at room temperature, followed by 3×5 min washes with PBS-T and visualized using the ECL reagent (Pierce) and exposure on medical X-ray films.

## Animals, fixation, and tissue processing

Mouse (C57BL6/C mice), rat (Sprague-Dawley), and monkey (*M. fascicularis*) brain sections were obtained from the brain bank of HM CINAC. All procedures with animals were carried out in accordance with the Directive of the Council of the European Communities (2010/63/EU) and Spanish

legislation (RD53/2013) on animal experimentation. Animal perfusion, fixation, and tissue processing were carried out as previously described (*Del Rey et al., 2021*; *Del Rey et al., 2022*).

## Brain immunohistochemistry

Coronal free-floating 30-mm-thick sections were washed with Tris buffer and treated with citrate buffer (pH 6) for 30 min at 37°C for antigen retrieval. Inhibition of endogenous peroxidase activity was carried out using a mixture of 10% methanol and 3% concentrated $H_2O_2$ for 20 min at room temperature. Normal serum of the same species as the secondary antibody (Normal Goat Serum [NGS]) was applied for 3 hr to block nonspecific binding sites. The sections were immunostained with anti-p97 antibody (Rabbit polyclonal anti-vcp, Sigma-Aldrich HPA012728, dilution 1: 1000) at 4°C for 72 hr. The sections were washed with Tris-buffered saline (TBS) and incubated for 2 hr with a solution containing the secondary biotinylated antibody (goat anti-rabbit, dilution 1:400; Chemicon, Burlington, MA, USA) in TBS-NGS. The sections were then incubated for 45 min with the avidin-biotin-peroxidase complex (PK-6100, ABC Vectastain; Vector Laboratories, Burlingame, CA, USA). Immuno-histochemical reactions were visualized by incubating the sections with 0.05% 3,3-diaminobenzidine (DAB, Sigma, St. Louis, MO, USA) and 0.003% $H_2O_2$. The sections were then dehydrated with gradu-ated ethyl alcohol (EtOH) and rinsed in two xylene changes before being mounted in dibutyl polysty-rene xylene phthalate (DPX) and applying glass coverslips. Omission of the primary antibody resulted in a lack of staining (images not shown).

## Molecular modeling

An atomic model of UBX-Nb[(GFP)] was predicted using the AlphaFold2 algorithm via the ColabFold notebook (*Mirdita et al., 2021*; *Jumper et al., 2021*). The FAF1 UBX domain position on the p97 hexamer was modeled by LSQ superposition of the p97 N-domain portion of a p97-N/FAF1 UBX co-crystal structure (PDB 3qq8) with the N-domain of chain A of a p97 hexamer cryo-EM structure (PDB 7jy5) using the program Coot (*Hänzelmann et al., 2011*; *Pan et al., 2021*; *Emsley et al., 2010*). Subsequently, the FAF1 UBX domain portion of the AlphaFold prediction was aligned with the p97/ FAF1 UBX domain model using LSQ superpose in Coot. The conformation of the flexible linker between Nb[(GFP)] and FAF1 UBX domain in the fusion protein model was manually adjusted using the 'Regularize Zone' feature in Coot. Finally, the position of GFP was modeled by LSQ superposition of the nanobody portion of the adjusted UBX-Nb[(GFP)] coordinates with the crystal structure of a Nb[(GFP)]/ GFP co-crystal structure (PDB 3ogo).

## Statistical analysis

The results are presented as the mean ± SEM. All statistical analyses were performed using the GraphPad Prism software. For all western blots, an unpaired Student's t-test was performed to deter-mine differences between control cells transfected with an empty vector versus UBX-Nb[(GFP or Syn87)]-treated cells. A confidence level of 95% ($p<0.05$) was accepted as statistically significant.

## Acknowledgements

This work was funded by ANID MPG190011 to ARF; FONDECYT REGULAR 1200427 to ARF; FONIS EU-LAC T010047 to CSR, PCC, JO, JB, NLR, and ARF; to ARF; ANID-EXPLORACIÓN 13220075 to ARF and GVN; ANID 3220635 to GVN; MPG 190011 to ARF; Centro Ciencia and Vida, FB210008, Financiamiento Basal para Centros Científicos y Tecnológicos de Excelencia de ANID to ARF; ANID-STINT CS2018-7952. JB was funded by ISCIII Miguel Servet Program CP19/00200 and FIS PI20/00496. Graduate fellowship ANID N° 22170632 to CS; NLGR was funded by Becas Santander Iberoamerica Investigacion 2018/2019 and grant S2017/BMD-3700 (NEUROMETAB-CM) from Comunidad de Madrid; DS was supported by the German Research Foundation (DFG) Emmy Noether Programme SCHW1851/1-1. We would like to thank Anne Berking and Amber Philp for proofreading and Felipe Serrano for illustration.

# Additional information

### Competing interests

Constanza Salinas-Rebolledo, Alejandro Rojas-Fernandez: Is an investor of the patent request WO2023187761A1. The other authors declare that no competing interests exist.

### Funding

| Funder | Grant reference number | Author |
| --- | --- | --- |
| Fondo Nacional de Desarrollo Científico y Tecnológico | 1200427 | Alejandro Rojas-Fernandez |
| Fondo Nacional de Desarrollo Científico y Tecnológico | 3220635 | Guillermo Valenzuela-Nieto |
| Fondo Nacional de Desarrollo Científico y Tecnológico | 13220075 | Guillermo Valenzuela-Nieto<br>Alejandro Rojas-Fernandez |
| Agencia Nacional de Investigación y Desarrollo | 21170632 | Constanza Salinas-Rebolledo |
| Centro Ciencia ficos y Tecnológicos de Excelencia de ANID | FB210008 | Alejandro Rojas-Fernandez |
| Max Planck Society | 190011 | Alejandro Rojas-Fernandez |
| ISCIII Miguel Servet Program | CP19/00200 | Javier Blesa |
| ISCIII Miguel Servet Program | FIS PI20/00496 | Javier Blesa |
| Becas Santander Iberoamerica Investigacion 2018/2019 | S2017/BMD-3700 | Natalia López-González del Rey |
| German Research Foundation (DFG) Emmy Noether Programme | SCHW1851/1-1 | David Schwefel |
| FONIS EU-LAC | T010047 | Constanza Salinas-Rebolledo<br>Pedro Chana-Cuevas<br>José A Obeso<br>Javier Blesa<br>Natalia López-González del Rey<br>Alejandro Rojas-Fernandez |

The funders had no role in study design, data collection and interpretation, or the decision to submit the work for publication.

### Author contributions

Constanza Salinas-Rebolledo, Conceptualization, Resources, Data curation, Software, Formal analysis, Supervision, Funding acquisition, Validation, Investigation, Visualization, Methodology, Writing – original draft, Project administration, Writing – review and editing; Javier Blesa, Conceptualization, Resources, Data curation, Formal analysis, Funding acquisition, Investigation, Project administration, Writing – review and editing; Guillermo Valenzuela-Nieto, Resources, Formal analysis, Investigation, Writing – original draft; David Schwefel, Conceptualization, Resources, Data curation, Formal analysis, Validation, Investigation, Methodology, Writing – original draft, Writing – review and editing; Natalia López-González del Rey, Supervision, Investigation, Methodology, Writing – review and editing; Maxs Méndez-Ruette, Investigation, Writing – review and editing; Janine Burkhalter, Data curation, Formal analysis, Investigation, Writing – original draft, Writing – review and editing; Elizabeth Carrazana, Francisca Díaz-Tejeda, Ignacio Arias Catalán, Investigation, Methodology, Writing – review

and editing; Claudio Cappelli Leon, Natalia Salvadores, Investigation, Methodology, Project administration, Writing – review and editing; Luis Federico Bátiz, Data curation, Formal analysis, Validation, Investigation, Visualization, Methodology, Writing – original draft, Writing – review and editing; Ronald Jara, Data curation, Formal analysis, Supervision, Funding acquisition, Investigation, Visualization, Writing – original draft, Writing – review and editing; José A Obeso, Resources, Supervision, Funding acquisition, Investigation, Methodology, Writing – original draft, Writing – review and editing; Pedro Chana-Cuevas, Conceptualization, Resources, Formal analysis, Supervision, Funding acquisition, Investigation, Methodology, Writing – original draft, Project administration, Writing – review and editing; Gopal P Sapkota, Data curation, Supervision, Funding acquisition, Validation, Investigation, Visualization, Methodology, Writing – original draft, Writing – review and editing; Alejandro Rojas-Fernandez, Conceptualization, Resources, Data curation, Formal analysis, Supervision, Funding acquisition, Validation, Investigation, Visualization, Methodology, Writing – original draft, Project administration, Writing – review and editing

## Author ORCIDs
Constanza Salinas-Rebolledo (ORCID) https://orcid.org/0000-0002-6220-504X
Javier Blesa (ORCID) https://orcid.org/0000-0002-4257-1325
David Schwefel (ORCID) https://orcid.org/0000-0002-2945-0908
Ronald Jara (ORCID) https://orcid.org/0009-0005-2861-3911
Pedro Chana-Cuevas (ORCID) https://orcid.org/0000-0001-9946-3299
Gopal P Sapkota (ORCID) https://orcid.org/0000-0001-9931-3338
Alejandro Rojas-Fernandez (ORCID) https://orcid.org/0000-0002-6004-5425

## Decision letter and Author response
Decision letter https://doi.org/10.7554/eLife.101496.sa1
Author response https://doi.org/10.7554/eLife.101496.sa2

---

## Additional files

### Supplementary files
MDAR checklist

### Data availability
All data generated in this study are included in the manuscript and supporting files. Source data files for all gels and blots are provided for each main and figure supplement, including original unprocessed images and labeled versions.

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
