## [Editor Report]

This study describes a valuable PROTAC approach in which the UBX domain of FAF1 fused to a nanobody (Ubx-Nb) can target protein of interest for degradation. The authors provide convincing evidence, showing that Ubx-Nb with a nanobody recognizing GFP can reduce the cellular levels of several GFP-fusion model substrates including some aggregation-prone proteins relevant to neurodegenerative diseases. The study will be of broad interest to cell biologists in the targeted protein degradation field.

---

## [Decision Letter]

[Editors' note: this paper was reviewed by Review Commons.]

---

## [Author Response]

Reviewer #1 (Evidence, reproducibility and clarity):The paper by Salinas-Rebolledo et al. describes a novel PROTAC approach in which the UBX domain of FAF1 is fused to a nanobody that recognizes the target protein. The idea is that the UBX domain will bind the fusion protein to the p97 ATPase, a major ATPase involved in the unfolding of many proteins. The target protein recognized by the UBX-nanobody fusion (UBXNb) is then supposed to be unfolded in a ubiquitin-independent manner and subsequently degraded by the proteasome.The authors provide evidence that Ubx-Nb, containing a nanobody recognizing GFP, can colocalize with GFP fusion proteins in the nucleus and to liquid-liquid phase separation structures. Importantly, the fusion can reduce the cellular levels of the target proteins. The authors confirm that degradation triggered by Ubx-Nb is proteasome dependent. Ubx-Nb can also promote the degradation of model proteins that form aggregates relevant to neurodegenerative diseases.The major issue with the paper is that it does not provide mechanistic insight into the degradation mechanism. First, the data implicating p97 in degradation are conflicting. On the one hand, siRNA of p97 compromises degradation (although degradation is not completely inhibited; see Figure 4G), but on the other hand, an inhibitor of p97 does not have an effect. The authors have not shown that target proteins are actually unfolded by their artificial adaptor (in vitro experiments would be required).In addition, it would be important to show co-localization in vivo with p97.Thus, the role of p97 is not convincingly established. Another major question is how the unfolded, non-ubiquitinated proteins would be degraded by the 26S proteasome. Is there a ubiquitin ligase required after substrate unfolding?Overall, the paper reports some intriguing effects of their designed PROTAC adaptor, but the mechanism by which it functions remains unclear. The findings of the manuscript appears too preliminary in its current version for it to be of value to the community.

We appreciate the feedback provided and acknowledge the reviewers' concerns regarding the mechanistic insight into the degradation process. Below, we address some of the point raised:

We have been working on a way to demonstrate that our technology is capable of reducing aggregate degradation using cell models available in our laboratory, and we came up with the idea of using an aggregate amplification assay with cell models. We have successfully demonstrated the ability to induce aggregate formation, and furthermore, it has also been demonstrated that fusion of UBX with GFP nanobodies is capable of reducing intracellular active aggregates of wild-type GFP-fused alphasynuclein, Figure 6E.

We acknowledge that our current data do not fully elucidate the degradation mechanism involving p97, however we unveil a practical phenomenon with potential clinical associated uses.

Regarding the ablation of p97 siRNA we do observe significant rescue of Emerin in the Figure 4G. Even more we also observed a slight increase in the levels of the UBX fused to the nanobody anti GFP indicating that even the levels of the synthetic construct are down regulated by its association to p97.

We agree that the results derivate of the used of p97 inhibitor are intriguing, regarding this matter we can only pointed that the selective inhibitor used in our study binds to the D2 domains of p97, as reported by Zhou et al. (J Med Chem. 2015). There are molecules that bind to the D1 domain and enhance p97-mediated degradation (Figuerola-Conchas et al., ACS Chem Biol. 2020). The primary ATPase function of p97 is associated with the D2 domain, but ATP binding to the D1 domain is crucial for hexamer formation and N-terminal domain conformation, which regulates cofactor binding and various functions of p97. Therefore, it is possible that our p97-PROTAC activates the D1 domain or interacts with cofactors enhancing protein degradation mediated by p97.

In order to demonstrate that proximity is sufficient for p97-mediated degradation, we performed an experiment in which a GFP-tagged p97 was overexpressed at increasing concentration and a single myc-tagged Nanobody against GFP was co-transfected. We demonstrated that the levels of the myc-tagged nanobody also decreased along the increasing concentration of GFP-tagged p97. We suggest that this experiment also supports the idea that proximity to p97 is mechanistically the most relevant feature for the p97-mediated PROTAC we presented in this manuscript (Supplemental Figure 1H and 1I).

We added a co-localization profiles into the supplemental Figure 1A, 1B and 1C between the targets protein and the p97-PROTAC. However, the in vivo application feld out of the scope of this manuscript.

In order to demonstrate that the p97-PROTAC-mediated degradation occurs in a ubiquitin-independent manner, we used the Ubiquitin E1 inhibitor Pyr-41. We selected p53 as a model, which is a constitutively degraded protein in Hela cells by the viral complex E6AP/E6. We determined the concentration at which ubiquitination was decreased to the point where p53 accumulation was observed (Supplemental Figure 1G). Further, a similar concentration of the inhibitor was used in a p97-mediated degradation assay in Figure 4G. Since we did not observe the inhibitory effect on the degradation, we concluded that this is a process independent of ubiquitination, either before or after the protein could be unfolded by p97.

Previous evidence suggests that some other protein can be directed to the proteasome via both ubiquitin-dependent and ubiquitin-independent mechanisms. For instance, Butler et al. (2016) demonstrated that fusing NbSyn87 with the mouse Ornithine Decarboxylase (ODC) PEST degron effectively reduced protein levels and this reduction was achieved by harnessing the innate cellular machinery responsible for ubiquitin-independent proteolysis.

Makaros Y, Raiff A, Timms RT, Wagh AR, Gueta MI, Bekturova A, Guez-Haddad J, Brodsky S, Opatowsky Y, Glickman MH, Elledge SJ, Koren I. Ubiquitin-independent proteasomal degradation driven by C-degron pathways. Mol Cell. 2023 Jun 1;83(11):1921-1935.e7. doi: 10.1016/j.molcel.2023.04.023. Epub 2023 May 17. PMID: 37201526; PMCID: PMC10237035.Butler DC, Joshi SN, Genst E, Baghel AS, Dobson CM, Messer A. Bifunctional Anti-NonAmyloid Component α-Synuclein Nanobodies Are Protective In Situ. PLoS One. 2016 Nov 8;11(11):e0165964. doi: 10.1371/journal.pone.0165964. PMID: 27824888; PMCID: PMC5100967.Erales J, Coffino P. Ubiquitin-independent proteasomal degradation. Biochim Biophys Acta. 2014 Jan;1843(1):216-21. doi: 10.1016/j.bbamcr.2013.05.008. Epub 2013 May 14. PMID: 23684952; PMCID: PMC3770795.Donghong Ju, Youming Xie, Proteasomal Degradation of RPN4 via Two Distinct Mechanisms, Ubiquitin-dependent and -independent*, Journal of Biological Chemistry, Volume 279, Issue 23, 2004, Pages 23851-23854, ISSN 0021-9258.

We also speculate that the degradation may also occur via the autophagy-lysosome pathway. These speculations will be added to the Discussion section, and we plan to investigate this pathway in detail in a subsequent scientific article focused on the mechanism of action of p97-PROTAC.

We believe that our findings present an innovative and unique tool, we appreciate the reviewers' comments and hope that our detailed response and future research plans address their concerns.

Minor points:Figure 1B: Although emerin is reported to be a nuclear envelope protein, it is not localized to the NE, but throughout the ER, likely because the protein was too highly expressed.

We agree with the reviewer that overexpression can lead to undesirable effects, in this case low transfection levels (0.5 ug DNA) were used, and the localization was observed in the nuclear membrane and ER, as reported in several cell lines even under endogenous levels: https://www.proteinatlas.org/ENSG00000102119-EMD/subcellular. To further support this observation, we have included an additional reference in the manuscript.

Figure 1B: ETV co-localization is not obvious from the figure.

We have included histograms for the colocalization in the supplemental Figure 1A, 1B and 1C to highlight the co-localization reported

Figure 4: The depletion of p97 leads to cell death, so it is unclear whether the siRNA effect is specific.

We appreciate your comment and would like to clarify that there are published studies where the p97 gene has been silenced in HeLa cells without causing complete cell death for an specific time frame, we also like to highlight that siRNA ablation in combination with western blot analysis is a very sensitive technique but it´s not a demonstration of a full depletion as it would be the generation of Knock out cellular models.

Reported studies

Wójcik C, Yano M, DeMartino GN. RNA interference of valosin-containing protein (VCP/p97) reveals multiple cellular roles linked to ubiquitin/proteasome-dependent proteolysis. J Cell Sci. 2004 Jan 15;117(Pt 2):281-92. doi: 10.1242/jcs.00841. Epub 2003 Dec 2. PMID: 14657277.Beskow A, Grimberg KB, Bott LC, Salomons FA, Dantuma NP, Young P. A conserved unfoldase activity for the p97 AAA-ATPase in proteasomal degradation. J Mol Biol. 2009 Dec 11;394(4):732-46. doi: 10.1016/j.jmb.2009.09.050. Epub 2009 Sep 24. PMID: 19782090.Yahiro K, Tsutsuki H, Ogura K, Nagasawa S, Moss J, Noda M. Regulation of subtilase cytotoxin-induced cell death by an RNA-dependent protein kinase-like endoplasmic reticulum kinase-dependent proteasome pathway in HeLa cells. Infect Immun. 2012 May;80(5):1803-14. doi: 10.1128/IAI.06164-11. Epub 2012 Feb 21. PMID: 22354021; PMCID: PMC3347452.

Figure 5C: The colocalization of GFP-HTT Q23 with UBX-Nb(GFP) is not entirely convincing.

In this particular case the GFP-HTT Q23 is distribute in the cytoplasm and it´s does not accumulates in structures that could reference as references to high light the colocalization.

Reviewer #1 (Significance):Overall, the paper reports some intriguing effects of their designed PROTAC adaptor, but the mechanism by which it functions remains unclear. The findings of the manuscript appears too preliminary in its current version for it to be of value to the community.Reviewer #2 (Evidence, reproducibility and clarity (Required)):Summary:The authors have developed a p97-directed proteolysis-targeting chimera (PROTAC) that operates independently of ubiquitin. This system employs a camelid nanobody to selectively recognize target proteins, tethered to p97 through the UBX domain of the p97 adapter FAF1. The anti-GFP nanobody effectively targets various GFP-fusion proteins for degradation via the proteasome, relying on p97 for its mechanism of action. The authors validate the presence of p97 in brain tissues of Non-Human Primates (Nhp) Macaca fascicularis, rat (Sprague Dawley), and mouse (C57BL6/C), supported by proteasome inhibition and p97 RNA silencing data. Importantly, the p97-PROTAC mechanism operates independently of ubiquitination, demonstrated through degradation of clinically relevant proteins such as α-synuclein using a camelid nanobody (NbSyn87).Major comments:Anti-GFP Nanobody clarification: Details about the original anti-GFP nanobody are unclear, which makes reproducing the current work a challenge for outside labs.20. Fulcher LJ, Macartney T, Bozatzi P, Hornberger A, Rojas-Fernandez A, Sapkota GP. An affinity-directed protein missile system for targeted proteolysis. Open Biol. Oct 2016;6(10)doi:10.1098/rsob.160255 o I assume that the anti-GFP nanobody is aGFP from supplementary figure #2 in the Fulcher manuscript but is unclear as they also have used anti-GFP nanobody aGFP16.Amino acids from FAF1-UBX domain: Further clarity is needed regarding the amino acid details from the FAF1-UBX domain, which may have been disclosed in a patent application but should be explicitly outlined in the methods section.

We appreciate the comment regarding the need for further clarity on the amino acid details from the FAF1-UBX domain. To address this, we have added a supplementary figure which includes a table outlining the amino acid sequences of our construct and the FAF1-UBX domain. This supplementary figure 1 provides a detailed representation of the amino acid sequences. We hope this addition meets your requirements and provides the necessary information for a comprehensive understanding of our work. We also cited the studies from Fulcher et all accordingly.

Degradation of Proteins of Clinical Interest: The data presented is not convincing enough to support the stated claims that the PROTAC is clearing aggregated mutant HTT. In Figure 5, there is an abundance of GFP-HTT Q74 puncta. While the western blot data suggests a reduction of soluble GFP-HTT-Q74 protein levels, it does not account for aggregated HTT. Aggregated HTT does not efficiently enter the separating gel during electrophoresis. To make these claims the authors need to 1. Show the level of mHTT Q74 aggregation in the empty control groups so that a comparison can be visually made between empty control groups and UBX-Nb(GFP) treated groups. A similar comparison would be useful with the GFP-HTT Q23 treated cells as well.Visualization of Aggregated Proteins: The continued visibility of puncta raises doubts about the system's efficacy in degrading aggregated proteins. Including comparisons between untreated and treated cells for all test systems would strengthen the argument. It would be useful to show a comparison between the untreated controls and UBX-Nb (GFP) treated cells for all the test systems shown.

We acknowledge that the data presented in Figure 5 may not be sufficient to support the claim that the PROTAC is clearing aggregated mutant HTT. The western blot data suggests a reduction in soluble GFP-HTT-Q74 protein levels, but it does not account for aggregated HTT, which does not efficiently enter the separating gel during electrophoresis.

To address this, we conducted additional experiments using Q24-Htt, where we observed the aggregates of huntingtin and the reduction of high molecular weight aggregates upon treatment with UBX-Nb (Supplemental Figure-2B). We also attempt and spend several months implementing the RT-QuIC system to determine the reduction of seeds and aggregates in cells, however the experimental setting is very variable and not fully reproducible at this stage. Regardless our strong commitment to directly demonstrating the reduction of the active seeds and toxic aggregates remain to be solved. We also believe that in vitro reconstitution could be a direct manner to measure the effect of the p97-PROTAC. However, unfortunately, this approach fell outside of the scope of this study.

To demonstrate the reduction of aggregates in tissue culture, we adapted the RealTime Quaking-Induced Conversion (RT-QuIC) assay, a highly sensitive method for amplifying α-synuclein aggregates to detect and quantify intracellular aggregates. Initially, we confirmed that transient transfection increases aggregate levels, which can be directly measured using this method. We then assessed the effect of UBX fused to Nanobody 87 and UBX fused to a GFP-targeting nanobody on aggregate seeding. While UBX-Nanobody 87 showed a tendency to reduce aggregate seeding, UBX-GFP nanobody significantly decreased intracellular aggregate levels. This assay required several months of optimization, but we are pleased to present a useful adaptation of the RT-QuIC method along with evidence that UBX-nanobody fusions can effectively reduce protein aggregation.

We additionally demonstrate that our technology is capable of reducing aggregate degradation using cell models available for the induction of intracellular aggregates of α synuclein and associated to aggregates amplification assay with cell models. We have successfully demonstrated the ability to induce aggregate formation in cells, and furthermore, it has also been demonstrated that fusion of UBX with GFP nanobodies is capable of reducing intracellular active aggregates of wild-type GFP-fused alphasynuclein, Figure 6E.

Minor comments:The In-text citations should be placed outside of the sentence. Example: The ubiquitin proteasome system (UPS) regulates protein abundance by specific E3 ubiquitin ligases, which catalyze ubiquitin chain formation on the substrates, inducing their proteasome mediated degradation. 1-4Sentence two in the introduction is missing a period. It is unclear whether sentence three is a heading or part of paragraph one.There are additional formatting issues. It would be easier to read the paper if there was a space between paragraphs. Page numbers would be helpful.Page 2. Missing word. Inclusion body myopathy associated with Paget's disease.Figure 1. D, F, and E. Missing annotation to denote significance.Figure 2G Missing annotation to denote significance.Figure 4. Missing annotation to denote significance for figures 4D, 4F, 4H.Is Figure 4I significantly different or not? In Figure 6, you use ns to denote not significant. This feels like it is an important point that you would want to make that the effect is dependent on p97. When you knock out p97 the degradation capacity of UBX-Nb is lost.

We appreciate your careful review and constructive feedback. We have addressed all the minor comments as follows:

Citations: We have revised the placement of citations throughout the manuscript to ensure they are positioned outside the sentence.

Introduction and Formatting: We have corrected the introduction and improved the manuscript’s readability by adding spaces between paragraphs and including page numbers for better navigation.

Page 4: We have corrected the missing word in "Inclusion body myopathy associated with Paget's disease of bone and Frontotemporal Dementia (IBMPFD)."

Figure Annotations: We have added the missing significance annotations in Figures 1D, 1F, and 1E. We have included the appropriate significance annotations in Figure 2G. We have revised Figure 4 to ensure significance annotations are present for Figures 4D, 4F, and 4H. Also, we have clarified that Figure 4H shows a significant difference, emphasizing that when p97 is knocked out, the degradation capacity of UBX-Nb is lost. We have ensured that the data presentation clearly reflects the dependence on p97. And finally, for Figure 4I, we have used "ns" to denote non-significance.

Reviewer #2 (Significance):The p97-PROTAC system is an ubiquitin-independent approach to degrade intracellular proteins. This system was able to target proteins for degradation at diverse subcellular locations, integral membrane protein residing at the inner nuclear membrane, chromatin located, and liquid-liquid phase separated compartments. The ability to clear alphasynuclein builds on previous research suggesting that ubiquitin-independent degradation of α-synuclein could be a therapeutic approach to treat synucleinopathies such as Parkinson's Disease. However, the ability of this approach to clear aggregated proteins is not convincing, given the presence of visible aggregates in the treated cells.The investigations with NbSyn87 build upon prior research by Butler et al. (2016), who fused NbSyn87 with the mouse Ornithine decarboxylase (ODC) PEST degron. This fusion strategy not only facilitates the targeting of α-synuclein but also harnesses the innate cellular machinery responsible for ubiquitin-independent proteolysis. The current approach demonstrates and alternative mechanism to direct α-synuclein (and other proteins) into the proteasome for ubiquitin-independent clearance.At the current state of development, this research is of interest to specialized audience with antibody engineering backgrounds; however, it holds translational potential for clearance of toxic proteins.My research interests is in the development of therapeutics for the treatment of neurodegenerative diseases including Huntington's Disease, Parkinson's Disease, and Alzheimer's Disease.